# Efficacy and Safety of Denosumab Therapy for Osteogenesis Imperfecta Patients with Osteoporosis—Case Series

**DOI:** 10.3390/jcm7120479

**Published:** 2018-11-24

**Authors:** Tsukasa Kobayashi, Yukio Nakamura, Takako Suzuki, Tomomi Yamaguchi, Ryojun Takeda, Masaki Takagi, Tomonobu Hasegawa, Tomoki Kosho, Hiroyuki Kato

**Affiliations:** 1Department of Orthopaedic Surgery, Shinshu University School of Medicine, Matsumoto, Nagano 390-8621, Japan; 16m0042e@shinshu-u.ac.jp (T.K.); takako1119@shinshu-u.ac.jp (T.S.); hirokato@shinshu-u.ac.jp (H.K.); 2Center for Medical Genetics, Shinshu University Hospital, Matsumoto, Nagano 390-8621, Japan; t_yamaguchi@shinshu-u.ac.jp (T.Y.); ktomoki@shinshu-u.ac.jp (T.K.); 3Division of Medical Genetics, Nagano Children’s Hospital, Azumino, Nagano 399-8288, Japan; ryojun_takeda@icloud.com; 4Department of Pediatrics, Keio University School of Medicine, Tokyo 160-8582, Japan; mtakagi1027@hotmail.com (M.T.); thaseg@keio.jp (T.H.); 5Department of Medical Genetics, Shinshu University School of Medicine, Matsumoto, Nagano 390-8621, Japan

**Keywords:** bone mineral density, denosumab, fractures, osteogenesis imperfecta, osteoporosis

## Abstract

Osteogenesis imperfecta (OI) is a connective tissue disorder that is characterized by low bone density leading to recurrent fractures. The efficacy of the anti-resorption drug denosumab for OI with osteoporosis is still largely unknown. We herein describe the clinical outcomes of eight osteoporotic cases of OI to examine the effects and safety of denosumab. This retrospective, consecutive case series included eight patients respectively aged 42, 40, 14, 22, 3, 51, 37, and 9 years. We measured the bone mineral density (BMD) of the lumbar 1–4 spine (L-BMD) and bilateral hips (H-BMD), bone-specific alkaline phosphatase, urinary type I collagen amino-terminal telopeptide, and tartrate-resistant acid phosphatase 5b before and during denosumab therapy. Despite multiple pretreatment fractures in the cohort, no fractures or severe side effects, such as hypocalcemia, were observed during the observational period apart from a fracture in a young pediatric girl. Both L-BMD and H-BMD were increased by denosumab in seven of eight cases. Bone turnover markers were inhibited in most cases by denosumab therapy. Denosumab treatment could generally raise BMD without any adverse effects. The agent therefore represents a good therapeutic option for OI with osteoporosis.

## 1. Introduction

Osteogenesis imperfect (OI) is a connective tissue disorder that is characterized by low bone density, recurrent fractures, and a wide genotypic and phenotypic spectrum of impaired collagen type I production [1]. OI is a rare disorder (1 in 15–20,000 births) and features skeletal fragility and substantial growth deficiency [2]. The clinical classification of OI was established in 1979 [3], which was later revised by van Dijk and Sillence in 2014 [4]. Among OI subtypes, type I is relatively mild and the most common. Type II OI causes death in the perinatal period, while type III induces obvious bone deformities and short stature due to frequent fractures [3,4].

Even in type I OI, it is important to treat bone fragility to reduce the risk of severe fractures that can diminish patient quality of life and activities of daily living. Intravenous bisphosphonate (BP) infusions for OI are the most broadly used medical treatment [5,6]. BPs can decrease long-bone fracture rates, but such injuries remain frequent [7]. In a clinical trial of the BP risedronate for OI, the mean increase in lumbar bone mineral density (L-BMD) at one year was 16.3%, although fractures occurred in 29 (31%) of 94 cases. At two years of therapy, fractures were recorded in 46 (53%) of 87 cases [8]. Thus, new antiresorptive drugs, such as denosumab, and anabolic agents are being investigated for OI with osteoporosis.

Denosumab is a fully human monoclonal antibody against receptor activator of nuclear factor-κB ligand, a mediator of osteoclastogenesis and osteoclast survival [9]. Subcutaneous injections of denosumab have been proposed as an alternative treatment approach for osteoporotic OI. Denosumab decreases bone resorption, increases BMD, and reduces fracture risk in postmenopausal women with osteoporosis [10]. We and others have recently reported that denosumab also improves bone fragility in children with OI and osteoporosis [11,12,13,14]. However, the precise efficacy and adverse effects of denosumab in OI are largely unknown, especially in pediatric cases.

This study retrospectively investigated eight patients with OI and osteoporosis to clarify the utility and safety of denosumab on bone fragility treatment. 

## 2. Materials and Methods

Seven consecutive female patients and a male patient diagnosed as having OI began treatment with denosumab for bone fragility. All patients had osteoporosis meeting the revised criteria established by the Japanese Society of Bone and Mineral Research [15], and most had experienced multiple fractures prior to denosumab therapy. Fracture event data were collected at every visit for each patient. The characteristics of the patients are summarized in Table 1. The frequency of fractures prior to treatment was 10 or more times in 6 patients (Patients 1, 2, 3, 4, 6, and 7), a single fracture in Patient 5 (the daughter of Patient 1), and no fracture in Patient 8 (the daughter of Patient 7). Denosumab of 60 mg was injected subcutaneously every 6 months into each patient (i.e., at 0, 6, 12, 18, 24, 30, 33, 39, 42, 49, and 54 months.). We also prescribed vitamin D supplementation tablets (762.5 mg of precipitated calcium carbonate, 200 IU of cholecalciferol, 59.2 mg of magnesium carbonate) twice daily to Patients 2, 3, and 4 and active vitamin D in the form of alfacalcidol to Patient 8 during denosumab administration.

### 2.1. Analysis of BMD

All subjects underwent dual-energy X-ray absorption (DXA) fan-beam bone densitometry (Lunar Prodigy; GE Healthcare Bio-Sciences Corp., Piscataway, NJ) at the lumbar 1–4 levels of the posteroanterior spine (L-BMD) (g/cm^2^) and total hips (H-BMD) (g/cm^2^) before and during denosumab administration. Timing period of BMD measurement after the start of denosumab therapy in each patient was approximately unified throughout the study period. 

### 2.2. Analysis of Laboratory Data

Serum calcium was measured by the Arsenazo reaction and phosphorus was measured by the molybdate reaction. As a bone formation marker, bone alkaline phosphatase (BAP) (μg/L) was measured by a chemiluminescent enzyme immunoassay. As bone resorption markers, urinary N-terminal telopeptide of type-I collagen (NTX) (nmol BCE/ mmol Cr) and tartrate-resistant acid phosphatase 5b (TRACP-5b) (mU/dL) were assessed by an enzyme-linked immunosorbent assay. Values of 1-alpha, 25-dihydroxyvitamin D3 (1,25[OH]_2_D) (ng/mL) and parathyroid hormone (PTH) (pg/mL) were determined by immunoradiometric assays. After overnight fasting and omission of the first morning samples, serum and urine specimens were collected between 8:30 a.m. and 11:00 a.m. Immunoassays were performed by SRL, Inc. (Tokyo, Japan). BMD and laboratory data were evaluated before, between 2 and 4 months, and at 6, 12, 18, and 24 months of denosumab administration unless otherwise indicated.

### 2.3. Ethical Approval

This investigation was approved by the Institutional Ethical Review Board of Shinshu University School of Medicine, Japan, prior to its initiation. Written informed consent was obtained from all subjects and the mother of the 3-year-old patient. Study methods were carried out in accordance with the approved guidelines.

## 3. Results

OI type according to Sillence’s classification was type Ib in Patients 1 and 5, type I similar in Patients 2 and 3, and type Ia in the remaining patients (Table 1), and was characterized by blue sclerae, premature deafness, mild to moderate bone fragility, and normal teeth. Patient 6 suffered from deafness in our cohort.

One incident of fracture, in Patient 5, was recorded at 11 months after the start of denosumab therapy (Table 1). Both L-BMD and H-BMD became increased in seven of eight cases (Figure 1). Serum albumin-corrected calcium levels during denosumab administration were stable in this cohort, with no hypercalcemia representing rebound inhibition of bone resorption (Table 2). No severe adverse effects were observed in this study.

The radiological and laboratory findings are summarized in Table 3 and Table 4. BMD generally increased in all patients. In Patients 2, 3, and 8, bone metabolism was strongly inhibited by denosumab treatment and 1,25 (OH)_2_D was sharply increased at 1 month of administration.

Genetic analysis revealed a missense mutation in the *COL1A1* gene encoding the collagen type I alpha 1 chain in Patients 1 and 5, a missense mutation in the *COL1A2* gene encoding the collagen type I alpha 2 chain in Patient 4, and a nonsense mutation in the *COL1A1* gene in Patients 6, 7, and 8. No mutations were detected by next generation sequencing of OI responsibility gene regions, including *COL1A1*, *COL1A2, SERPINF1*, and *IFITM5* in Patients 2 and 3. Their detailed clinical and molecular information will be described in another series (Table 1).

## 4. Case Highlights

The changes in BMD (Figure 1), laboratory findings (Figure 2), and bone markers (Figure 3) were plotted for each patient. In Patient 1, BAP, TRACP-5b, and NTX all transiently increased while 1,25 (OH)_2_D temporarily decreased. In Patient 2, 1,25 (OH)_2_D became increased at one month and gradually decreased. BAP was decreased for 18 months and increased thereafter. Urinary NTX was mostly maintained and TRACP-5b was decreased at one month. Both L-BMD and H-BMD increased during observation. In Patient 3, 1,25 (OH)_2_D became increased at one month and gradually decreased. BAP was decreased, while urinary NTX was increased at six and 30 months. TRACP-5b was decreased at one month but fluctuated thereafter. Both L-BMD and H-BMD increased throughout treatment. In Patient 4, serum albumin-corrected calcium fluctuated; 1,25 (OH)_2_D was increased for three months, and then decreased thereafter. Serum whole PTH was generally increased for 12 months. BAP was continuously decreased. Urinary NTX was decreased at one month, and then gradually increased. TRACP-5b became persistently decreased at one month. L-BMD gradually increased. In Patient 5, serum albumin-corrected calcium was decreased at one month but increased thereafter; 1,25 (OH)_2_D, whole PTH, BAP, and urinary NTX were persistently decreased. There were no remarkable changes in TRACP-5b. Serum albumin-corrected calcium was mildly increased; 1,25 (OH)_2_D was increased at one month and decreased thereafter. Serum whole PTH was unaffected for six months but increased at 12 months. BAP decreased for three months and stabilized thereafter. Urinary NTX was decreased at one month, and then fluctuated around baseline. TRACP-5b was greatly decreased. L-BMD was decreased at 12 months, while H-BMD was increased. In patient 6, serum albumin-corrected calcium was decreased but 1,25 (OH)_2_D was increased at one month. BAP, urinary NTX, and TRACP-5b were decreased for 12 months. L-BMD was decreased at 12 months but H-BMD was increased for 12 months. In Patient 7, 1,25 (OH)_2_D was mildly increased. Serum whole PTH was increased for three months. BAP was continuously decreased. Urinary NTX and TRACP-5b were markedly decreased at one month. L-BMD and H-BMD increased. In Patient 8, serum albumin-corrected calcium was greatly decreased at one month; 1,25 (OH)_2_D was markedly increased at one month and decreased thereafter. Serum whole PTH spiked at one month, and then decreased. BAP and urinary NTX were decreased at one month and increased thereafter. TRACP-5b was greatly decreased at one month but then stabilized. L-BMD and H-BMD increased.

## 5. Discussion

This case series including long-term denosumab therapy (over 30 months in five cases) for osteoporotic OI is the first of its kind in Japan. Since Patients 1, 2, 5, 6, and 8 had suffered repeated fractures prior to the start of this study and refused BP therapy due to reflux esophagitis, denosumab was selected as the first therapeutic modality. Patients 3, 4, and 7 had previously been treated with BPs and switched to denosumab due to a poor response. BMD was increased overall, and no fractures were observed in seven of eight patients. Patient 5 experienced a proximal femoral neck fracture during the observation period despite remarkable gains in BMD. Since she was an active 3-year-old girl, the risk of fracture in pediatric cases might be higher than in older OI patients [16]. 

As osteoporosis complicates many cases of OI, adequate and appropriate treatment is required for this condition. The low BMD typically present in OI is considered to be caused by high bone resorption [4]. Most patients with severe clinical OI symptoms are treated with pamidronate, risedronate, and other intravenous BPs [5,6]. However, poor responses to these drugs have been reported [7], necessitating alternative treatments for OI with osteoporosis, especially to prevent fragility fractures. OI-related fractures chiefly occur between childhood and puberty [16]. Patient 3’s height increased from 159 cm to 161 cm at two years of therapy, indicating that her growth was unaffected by denosumab. It was noteworthy that she had suffered multiple fractures in spite of earlier BP administration. The remaining patients had also experienced repeated fractures despite reaching adulthood. Since no fractures were noted for seven of eight patients during denosumab treatment, the drug appears to be effective for OI patients at high fracture risk. 

In our series, BMD increased soon after denosumab administration and remained improved during observation in all cases except Patient 6. She was post-menopausal at denosumab commencement, which might have explained why her L-BMD did not increase but her H-BMD did. The mechanism for this discrepancy is unknown, but may become clear after longer follow-up. Patients 2 and 3 had relatively high baseline BMD values but also repeated fractures prior to treatment. Thus, the modest increases in BMD in our cohort were thought to have partly resulted from originally high BMD. Overall, however, denosumab may improve BMD and decrease fracture risk in OI patients with osteoporosis.

Bone formation and bone resorption markers were both generally improved by denosumab in Patients 2 and 3, which was in line with other reports [11,17,18]. Patients 4, 5, 6, 7, and 8 also showed improvements. The causes of elevated BAP and TRACP-5b in Patient 1 are currently unknown. In most cases (Patients 3, 4, 6, 7, and 8), serum albumin-corrected calcium levels were markedly decreased at one month, while serum 1,25 (OH)_2_D levels were greatly increased at one month. Serum whole PTH was remarkably higher in Patients 3 and 8 at one month of treatment but relatively constant in the remaining cohort. Serum urinary NTX levels were decreased in most cases at one month, although Patient 3 showed a spike at 30 months by yet unknown mechanisms. Taken together, bone formation and bone resorption markers improved, serum albumin-corrected calcium was increased at one month, and serum 1,25(OH)_2_D and urinary NTX were greatly decreased at 1 month by denosumab administration.

Bone metabolism differs between adults and children [9]. L-BMD and H-BMD were both increased in pediatric Patients 5 and 8. In a related report, there was a significant increase in L-BMD percent change after 48 weeks of denosumab with no alterations in mobility parameters or serious adverse events [14]. Thus, denosumab therapy during childhood might be more beneficial since BMD increases tended to be higher than those in the adult OI patients. 

We previously reported on the effects of denosumab on growth in children [11]. Although the observational period was short, this was the first study on the results of denosumab for OI with osteoporosis. In our adolescent Patient 3, no apparent growth disturbance was observed as her height increased from 159 cm at the first administration to 161 cm at two years and she was within the normal age-based height range. The height of Patient 5 was 90.2 cm prior to treatment and 96.3 cm at one year. In Patient 8, it was 111.7 cm prior to therapy and 125.8 cm at one year. These findings support that denosumab can be considered for pediatric cases from the viewpoint of no growth disruption.

The majority (>90%) of patients with OI have autosomal dominant variants in *COL1A1/COL1A2* that lead to defects in type 1 collagen. In our study, mutations in *COL1A1/COL1A2* were harbored by 6 of 8 patients. Hoyer-Kuhn et al. also described a heterozygous mutation in the coding region of the interferon-induced transmembrane 5 (IFITM5) (c.-14C  >  T) gene, an underlying cause of OI type V [14]. Jin et al. reviewed that OI type VI was caused by mutations in serpin peptidase inhibitor, clade F, member 1 (*SERPINF1*), the gene coding for pigment epithelium-derived factor (PEDF) [19].

In our cohort, Patients 2 and 3 exhibited no remarkable mutations using a next generation sequencer; there is a possibility that they possessed other gene mutations such as those listed above. 

Hypocalcemia was absent in all patients during therapy apart from Patient 8 at one month. The reasons for this finding are unknown since denosumab is considered to decrease serum calcium levels in the initial treatment period as a consequence of the inhibition of bone turnover. However, we earlier reported no significant changes in serum calcium in a denosumab monotherapy group and in a denosumab with vitamin D and calcium supplementation group during observation [20]. 

Perhaps most strikingly, our cohort had experienced multiple fractures before denosumab treatment, but none were recorded during the administration period in 7 of 8 patients. This suggested that denosumab exerted fracture preventative and BMD increasing effects in OI patients with osteoporosis. Since denosumab discontinuation causes a marked decrease in BMD [21,22], BPs and other alternative treatments are recommended after its cessation. 

### Limitations

The limitations of the current study include its limited size, lack of a control group, lack of 25-Hydroxyvitamin D measurements, short observation period for some patients, and retrospective design. Nonetheless, our case series provides evidence of good BMD response and fracture prevention by denosumab treatment for OI with osteoporosis. 

## 6. Conclusions

This study is the first long-term treatment case series of denosumab for osteoporotic OI patients in Japan. Denosumab is well tolerated and provides generally good responses in BMD and bone turnover markers along with possible fracture prevention in OI patients with osteoporosis, and thus represents a good treatment option for such cases.

## Figures and Tables

**Figure 1 jcm-07-00479-f001:**
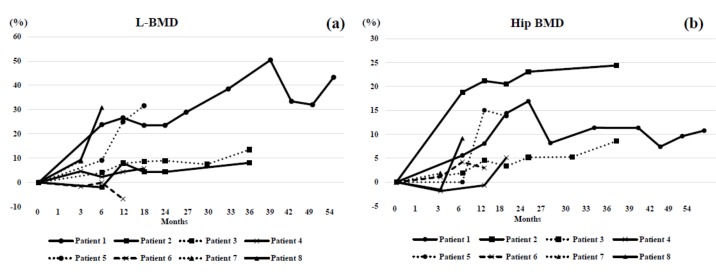
Changes in bone mineral density (BMD) before and during denosumab administration. (**a**) In most cases, the percent changes in lumbar 1–4 spine BMD (L-BMD) were positive during the follow-up period. In Patient 6, L-BMD was decreased at 12 months. In Patient 2, L-BMD was decreased until 6 months. (**b**) In most cases, the percent changes in bilateral hips BMD (H-BMD) were positive during observation. In Patient 4, H-BMD was decreased at 3 and 12 months. In Patient 8, H-BMD was decreased until 3 months. Patient 1: circles connected by a solid line. Patient 2: squares connected by a solid line. Patient 3: squares connected by a dotted line. Patient 4: cross marks connected by a solid line. Patient 5: circles connected by a dotted line. Patient 6: cross marks connected by a hashed line. Patient 7: triangles connected by a dotted line. Patient 8: triangles connected by a solid line. Circles, squares, and triangles represent family members.

**Figure 2 jcm-07-00479-f002:**
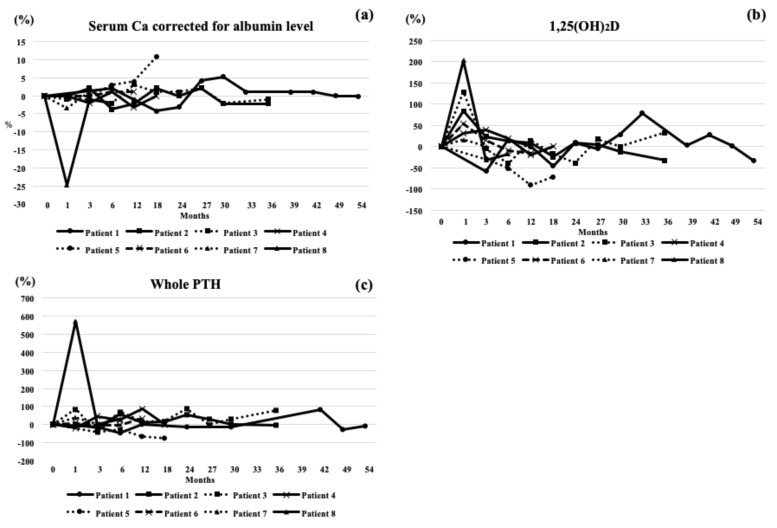
Changes in serum albumin-corrected calcium, 1,25 (OH)_2_D, and whole parathyroid hormone (PTH) levels before and during denosumab administration.(**a**) Percent changes in serum calcium were not remarkably altered during the follow-up period. Only Patient 6 showed hypocalcemia at 1 month, which returned to baseline values at 3 months of therapy. (**b**) In Patients 2, 5, 6, and 8, the percent changes in 1,25 (OH)_2_D were increased at 1 month but returned to baseline values thereafter. In other patients, they were not remarkably changed. (**c**) Percent changes in whole PTH did not change remarkably during follow-up. Only Patient 6 showed a notable decrease at 1 month, which returned to baseline values at 3 months of therapy. Patient 1: circles connected by a solid line. Patient 2: squares connected by a solid line. Patient 3: squares connected by a dotted line. Patient 4: cross marks connected by a solid line. Patient 5: circles connected by a dotted line. Patient 6: cross marks connected by a hashed line. Patient 7: triangles connected by a dotted line. Patient 8: triangles connected by a solid line. Circles, squares, and triangles represent family members.

**Figure 3 jcm-07-00479-f003:**
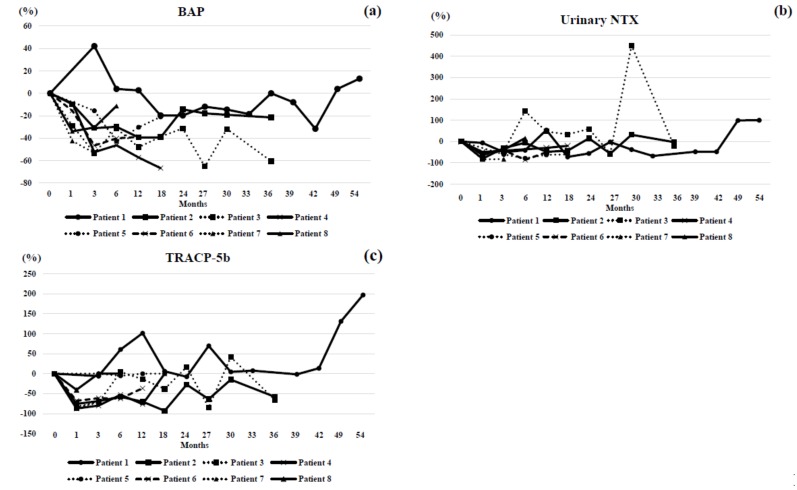
Changes in bone turnover markers before and during denosumab administration. (**a**) Only in Patient 1, the percent changes in bone alkaline phosphatase (BAP) were increased for the first 3 months, and then decreased or slightly increased thereafter. In other patients, they tended to be decreased during the observation period. (**b**) In Patient 3, the percent changes in urinary NTX were increased during 6 to 24 months, and then sharply increased at 30 months. In other patients, they were mostly constant or slightly inhibited during follow-up. (**c**) In Patient 1, the percent changes in TRACP-5b were increased at 6, 12, 27, 49, and 54 months. In other patients, they were mostly constant or slightly inhibited during the treatment period. Patient 1: circles connected by a solid line. Patient 2: squares connected by a solid line. Patient 3: squares connected by a dotted line. Patient 4: cross marks connected by a solid line. Patient 5: circles connected by a dotted line. Patient 6: cross marks connected by a hashed line. Patient 7: triangles connected by a dotted line. Patient 8: triangles connected by a solid line. Circles, squares, and triangles represent family members.

**Table 1 jcm-07-00479-t001:** Patient characteristics.

	Age, Years at Study Onset	Gender	OI Type	Height, cm at Study Onset	BMI at Study Onset	Previous Treatment	Occurrence of Fracture before Study	Occurrence of Fracture during Study	Causative Mutation	Dose, Frequency, and Duration of Denosumab Administration
**Patient 1**	42	F	Ib	138	24.1	None	>10	None	*COL1A1* c.G769A; p.G257R	60 mg every 6 months for 54 months
**Patient 2**	40	F	I similar	156	21.1	None	>10	None	Unidentified	60 mg every 6 months for 36 months
**Patient 3**	14	F	I similar	159	22.2	BP	>10	None	Unidentified	60 mg every 6 months for 36 months
**Patient 4**	22	F	Ia	134.5	23.6	BP	>10	None	*COL1A1* c.G1963C; p.G655R	60 mg every 6 months for 18 months
**Patient 5**	3	F	Ib	90.2	14.8	None	1	1	*COL1A1* c.G769A; p.G257R	15 mg every 6 months for 18 months
**Patient 6**	51	F	Ia	155	20.8	None	>10	None	*COL1A1* c.1243C>T; p.R415X	60 mg every 6 months for 12 months
**Patient 7**	37	M	Ia	166	25.4	BP	>10	None	*COL1A1* c.G607T; p.G203C	60 mg every 6 months for 4 months
**Patient 8**	9	F	Ia	111.7	15.2	None	None	None	*COL1A1* c.607G>T; p.G203C	30 mg every 6 months for 6 months

F: female, M: male, OI: osteogenesis imperfecta, BMI: body mass index, BP: bisphosphonate. Patient 5 is a daughter of Patient 1, Patient 3 is a daughter of Patient 2, and Patient 8 is a daughter of Patient 7.

**Table 2 jcm-07-00479-t002:** Patient laboratory data across observation periods for each patient.

	Serum Albumin-Corrected Ca Level (mg/dL)														
	Before (Reference Value: 8.5–10.2)	1M	2–4M	6M	12M	18M	24M	27M	30M	33M	36M	39M	42M	49M	54M
**Patient 1**	9.5			9.7	9.4	9.1	9.2	9.9	10	9.6	9.6	9.6	9.6	9.5	9.4
**Patient 2**	9.2	9.2	9.4	8.9	9.0	9.4	9.2	9.4	9.0		9.2				
**Patient 3**	9.6	9.5	9.5	9.4	9.9	9.7	9.7	9.8	9.4		9.5				
**Patient 4**	9.6	9.6	9.4	9.7	9.3	9.4									
**Patient 5**	10.2		10.2	10.5	10.6	11.4									
**Patient 6**	9.6	9.6	9.6	9.7	9.7										
**Patient 7**	9.0	8.7	9.1												
**Patient 8**	9.7	7.3	9.6	9.5											
	**Serum P (mg/dL)**														
	**Before (Reference value: 2.4–4.3)**	**1M**	**2–4M**	**6M**	**12M**	**18M**	**24M**	**27M**	**30M**	**33M**	**36M**	**39M**	**42M**	**49M**	**54M**
**Patient 1**	3.2		4.2	3.8	3.8	3.8	3.8	3.3	4.1	4.0		3.3	3.0	3.1	3.4
**Patient 2**	3.9	3.2	3.0	3.5	2.7	3.0	3.2	4.1	3.0		3.7				
**Patient 3**	3.8	3.1	3.8	3.5	3.8	3.9	3.5	4.2	4.6		3.7				
**Patient 4**	4.1	3.4	4.2	3.7	3.0	3.7									
**Patient 5**	5.3		5.1	5.6	6.0	5.8									
**Patient 6**	4.0	3.7	4.3	3.7	3.8										
**Patient 7**	2.8	2.6	2.3												
**Patient 8**	6.3	5.9	5.0	5.1											
	**Whole PTH (pg/dL)**														
	**Before (Reference Value: 8.3–38.7)**	**1M**	**2-4M**	**6M**	**12M**	**18M**	**24M**	**27M**	**30M**	**33M**	**36M**	**39M**	**42M**	**49M**	**54M**
**Patient 1**	40.6		35.5	21	40.6		35.3		35.4				73.5	29.3	37.6
**Patient 2**		17.1	13.9	27.4	19	19.6	26.1	22.3	17.3		16.3				
**Patient 3**	11.8	21.9	7.1	20.2	13.7	14	22.2	11.7	15.3		21.2				
**Patient 4**	14.8	12.2	21.5	18.5	27.5	21.7									
**Patient 5**	17.3		10.4	12.8	6.3	4									
**Patient 6**	22.8	24.5	22.2	22.5	30.3										
**Patient 7**	47.7	67.3	43.1												
**Patient 8**	16.9	114	19.2	22.1											

Ca: calcium, P: phosphorus, PTH: parathyroid hormone, M: months.

**Table 3 jcm-07-00479-t003:** Patient bone mineral density across observation periods for each patient.

	L-BMD (g/cm^2^)														
	Before	1M	2–4M	6M	12M	18M	24M	27M	30M	33M	36M	39M	42M	49M	54M
**Patient 1**	0.571			0.707	0.723	0.705	0.705	0.736		0.791		0.859	0.762	0.754	0.818
**Patient 2**	1.018			1.002	1.099	1.062	1.082				1.100				
**Patient 3**	1.070			1.113	1.154	1.162	1.166		1.15		1.215				
**Patient 4**	0.870		0.910	0.886	0.908	0.924									
**Patient 5**	0.396			0.432	0.495	0.521									
**Patient 6**	0.743		0.731	0.737	0.693										
**Patient 7**	1.224		1.332												
**Patient 8**	0.428		0.468	0.559											
	**L-BMD Z-score**														
	**Before**	**1M**	**2–4M**	**6M**	**12M**	**18M**	**24M**	**27M**	**30M**	**33M**	**36M**	**39M**	**42M**	**49M**	**54M**
**Patient 1**				−3.2	−3	−3.1	−2.9	−2.4		−2.8		−2.8	−2.9	−2.3	−2.7
**Patient 2**	−0.8			−1	−0.6	−0.4	−0.6		−0.4		−0.6				
**Patient 3**															
**Patient 4**	−1.6		−1.7	−1.9	−1.8	−1.6									
**Patient 5**															
**Patient 6**	−2.2		−2.2	−2.1	−2.4										
**Patient 7**	0.7		1.4												
**Patient 8**	−3.1		−2.6	−1.9											
	**L-BMD T-score**														
	**Before**	**1M**	**2–4M**	**6M**	**12M**	**18M**	**24M**	**27M**	**30M**	**33M**	**36M**	**39M**	**42M**	**49M**	**54M**
**Patient 1**				−3.4	−3.2	−3.4	−3.1	−27		−2.8		−2.8	−2.9	−2.4	−2.8
**Patient 2**	−0.8			−0.9	−0.4	−0.2	−0.6		−0.4		−0.6				
**Patient 3**															
**Patient 4**	−2.2		−1.7	−1.9	−1.8	−1.6									
**Patient 5**															
**Patient 6**	−2.9		−3	−3	−3.3										
**Patient 7**	0.7		1.4												
**Patient 8**															
	**H-BMD (g/cm^2^** **)**														
	**Before**	**1M**	**2–4M**	**6M**	**12M**	**18M**	**24M**	**27M**	**30M**	**33M**	**36M**	**39M**	**42M**	**49M**	**54M**
**Patient 1**	0.520			0.549	0.562	0.595	0.608	0.562		0.591		0.579	0.558	0.57	0.576
**Patient 2**	0.729			0.867	0.884	0.879	0.898				0.908				
**Patient 3**	0.946			0.964	0.988	0.977	0.993		0.996		1.027				
**Patient 4**	0.677		0.664		0.672	0.711									
**Patient 5**	0.409			0.370	0.470	0.465									
**Patient 6**	0.654		0.661	0.682	0.674										
**Patient 7**	0.979		0.997												
**Patient 8**	0.54		0.531	0.589											
	**H-BMD Z-score**														
	**Before**	**1M**	**2–4M**	**6M**	**12M**	**18M**	**24M**	**27M**	**30M**	**33M**	**36M**	**39M**	**42M**	**49M**	**54M**
**Patient 1**				−2.95	−2.85	−2.85	−2.85	−2.6		−2.45		−2.75	−2.7	−2.65	−2.55
**Patient 2**	−0.55			−0.45	−0.4	−0.25	−0.2		−0.15		−0.05				
**Patient 3**															
**Patient 4**	−2		−2.25	−2.25	−1.9	−2.25									
**Patient 5**															
**Patient 6**	−1.8		-1.75	-1.6	-1.6										
**Patient 7**	-0.25		-0.15												
**Patient 8**															
	**H-BMD T-score**														
	**Before**	**1M**	**2–4M**	**6M**	**12M**	**18M**	**24M**	**27M**	**30M**	**33M**	**36M**	**39M**	**42M**	**49M**	**54M**
**Patient 1**				-3.2	-3.1	-3.1	-3.1	-2.85		-3.15		-3.05	-3	-2.95	-2.9
**Patient 2**	−0.65			−0.55	−0.45	−0.25	−0.5		−0.4		−0.35				
**Patient 3**															
**Patient 4**	−2.15		−2.25	−2.25	−1.9	−2.25									
**Patient 5**															
**Patient 6**	−2.35		−2.3	−2.15	−2.2										
**Patient 7**	−0.45		−0.4												
**Patient 8**															

L-BMD: lumbar 1–4 bone mineral density, H-BMD: bilateral total hip bone mineral density, M: months.

**Table 4 jcm-07-00479-t004:** Patient laboratory data (bone metabolism markers) across observation periods for each patient.

	BAP (μg/L)														
	Before (Reference Value)	1M	2–4M	6M	12M	18M	24M	27M	30M	33M	36M	39M	42M	49M	54M
**Patient 1**	7.6 (2.9–14.5)		10.8	7.9	7.8	6.1	6	6.7	6.5	6.2	7.6	7	5.2	7.9	7.9
**Patient 2**	8.4 (2.9–14.5)	7.6	5.8	5.9	5.1	5.1	7.2	6.9	6.8		6.6				
**Patient 3**	16.7 (2.9–14.5)	11.9	7.8	11.5	8.7	6.9	11.5	5.8	11.4		6.6				
**Patient 4**	14.7 (2.9–14.5)	13.4	7	7.9	6.3	4.9									
**Patient 5**	62.2 (2.9–14.5)		52.6	35.6	43.4	60.9									
**Patient 6**	14.2 (2.9–14.5)	12	7.6	8.5	8.7										
**Patient 7**	20.3 (3.7–20.9)	11.7	9.5												
**Patient 8**	84.4 (2.9–14.5)	55.8	58.8	74.9											
	**Urinary NTX (nmol BCE/mmol Cr)**														
	**Before (Reference value)**	**1M**	**2–4M**	**6M**	**12M**	**18M**	**24M**	**27M**	**30M**	**33M**	**36M**	**39M**	**42M**	**49M**	**54M**
**Patient 1**	19.3 (9.3–54.3)	18.1	9.7	9.9	29.7	5.4	8.5	18.9	12	6.2		10.1	10.1	38.4	38.6
**Patient 2**	24.2 (9.3–54.3)	4.7	16.8	22.8	12.6	13.9	27.9	10.4	31.9		23.5				
**Patient 3**	32.9 (9.3–54.3)	12.9	18.1	79.7	48.4	44	52.3	14.4	181.2		25.8				
**Patient 4**	30.4 (9.3–54.3)	15.4			21.3	24.3									
**Patient 5**	1562.1 (9.3–54.3)		616.9	338	587.3	612.6									
**Patient 6**	35.4 (9.3–54.3)	13.2	21	5	17.5										
**Patient 7**	79.8 (13.0–66.2)	13.1	14												
**Patient 8**	518.2 (9.3–54.3)	207.9	320.3	601.8											
	**TRACP-5b (mU/dL** **)**														
	**Before** **(reference value)**	**1M**	**2–4M**	**6M**	**12M**	**18M**	**24M**	**27M**	**30M**	**33M**	**36M**	**39M**	**42M**	**49M**	**54M**
**Patient 1**	66 (120–420)		62	106	133	70	61	112	69	71		65	75	153	196
**Patient 2**	198 (120–420)	48	62	83	59	72	144	72	149		83				
**Patient 3**	414 (120–420)	60	114	432	356	256	484	64	586		142				
**Patient 4**	260 (120–420)	33	52	123	62	38									
**Patient 5**	1500 (120–420)		1500	1420	1500	1500									
**Patient 6**	319 (120–420)	103	123	120	203										
**Patient 7**	465 (170–590)	83	136												
**Patient 8**	1500 (120–420)	894	1500	1500											
	**1,25 (OH)_2_D (pg/mL)**														
	**Before** **(Reference Value)**	**1M**	**2–4M**	**6M**	**12M**	**18M**	**24M**	**27M**	**30M**	**33M**	**36M**	**39M**	**42M**	**49M**	**54M**
**Patient 1**	47.9 (20.0–60.0)		20.3	55.7	47.9	26.1	52.4	45.5	61.4	85.8		49.1	61.4	48.9	31.8
**Patient 2**	49 (20.0–60.0)	90.1	60.3	55.5	53.7	36.8	52.8	50.9	43.3		33.1				
**Patient 3**	57.2 (20.0–70.0)	131	54.1	34.7	65.6	47.3	35.2	67	47		75.9				
**Patient 4**	45.9 (20.0–60.0)	60.6	64.1	54.3	36.8	68.3									
**Patient 5**	58.9 (20.0–70.0)		41.3	29.2	5.6	10.7									
**Patient 6**	30.8 (20.0–60.0)	47.1	35.5	27.7	25.7										
**Patient 7**	69.8 (20.0–60.0)	81.6	83.8												
**Patient 8**	90.7 (20.0–70.0)	275	61.7	75.2											

BAP: bone-specific alkaline phosphatase, NTX: type I collagen amino-terminal telopeptide, TRACP-5b: tartrate-resistant acid phosphatase 5b, 1,25 (OH)_2_D: 1-alpha, M: months.

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
