# Peer review of "Efficacy and Safety of Denosumab Therapy for Osteogenesis Imperfecta Patients with Osteoporosis—Case Series"

_jcm, 2018, doi:10.3390/jcm7120479_

Round 1
Reviewer 1 Report
Article:
Efficacy and Safety of Denosumab Therapy for Osteogenesis Imperfecta Patients with Osteoporosis - Case Series
Tsukasa Kobayashi, Yukio Nakamura, Takako Suzuki, and Hiroyuki Kato
Department of Orthopaedic Surgery, Shinshu University School of Medicine, Matsumoto, Nagano, Japan.
Osteogenesis imperfect (OI) is a connective tissue disorder that is characterized by low bone mineral density (BMD) leading to bone fragility and recurrent fractures.
Kobayashi et al. investigated retrospectively, effects and safety of denosumab, an anti-resorptive agent, on bone fragility treatment in 8 osteoporotic cases of OI.
This study is the first of its kind in Japan. The authors conclude that denosumab represents a good therapeutic option for OI patients with osteoporosis.
My main criticism of the paper is:
Table 3:
For a better comparability BMD values should be presented additionally as Z- and T-scores, respectively.
Table 4:
Due to heterogeneity of age in the study cohort, reference values should be given for each age-group.
If possible, 25-Hydroxyvitamin D [25(OH)D] levels should be added to table 4. In contrast to 25(OH)D, circulating 1,25(OH)D is generally not a good indicator of vitamin D status [[1]]. 25(OH)D promotes calcium absorption in the gut and maintains adequate serum calcium and phosphate concentrations to enable normal bone mineralization.
There is one minor issue worthy of further consideration:
The mechanism of action of denosumab should be briefly described somewhere in the „Introduction“.
[1] Jones G. Pharmacokinetics of vitamin D toxicity. Am J Clin Nutr 2008;88:582S-6S.
Author Response
Editor-in-Chief
Journal of Clinical Medicine
November 12, 2018
Dear Editor-in-Chief,
Please find enclosed our revised manuscript, entitled “Efficacy and Safety of Denosumab Therapy for Osteogenesis Imperfecta Patients with Osteoporosis - Case Series”, which we would like to have re-considered for publication in Journal of Clinical Medicine. Although we were surprised with the sudden addition of another reviewer during the review process, we have addressed all of the comments and concerns raised by the Referees to the best of our ability.
Thank you again for your time and consideration.
Sincerely,
Yukio Nakamura
#Reviewer 1
(x) I don't feel qualified to judge about the English language and style
→ We have had the paper rechecked by a native English speaker. Several additions and deletions made in the editing process were not highlighted in the revised version.
My main criticism of the paper is:
Table 3:For a better comparability BMD values should be presented additionally as Z- and T-scores, respectively.
→Thank you for this advice. We have added them in Table 3.
Table 4:Due to heterogeneity of age in the study cohort, reference values should be given for each age-group.
→Thank you for this suggestion. We have added them in Table 2-4.
If possible, 25-Hydroxyvitamin D [25(OH)D] levels should be added to table 4. In contrast to 25(OH)D, circulating 1,25(OH)D is generally not a good indicator of vitamin D status [[1]]. 25(OH)D promotes calcium absorption in the gut and maintains adequate serum calcium and phosphate concentrations to enable normal bone mineralization.
[1] Jones G. Pharmacokinetics of vitamin D toxicity. Am J Clin Nutr 2008;88:582S-6S.
→ We absolutely agree with this comment. In Japan, however, only a single measurement of 25(OH)D is allowed for each patient during osteoporosis treatment, so we could not include these data in the present study. We will add such measurements in the future. We have stated this point as a limitation.
There is one minor issue worthy of further consideration:
The mechanism of action of denosumab should be briefly described somewhere in the Introduction.
→Thank you for this suggestion. We have added this point it in the text with a reference.
#Reviewer 2
( ) I don't feel qualified to judge about the English language and style
→ We have had the paper rechecked by a native English speaker. Several additions and deletions made in the editing process were not highlighted in the revised version.
Specific comments:
The duration, dose and frequency of Denosumab administration should be specified for each patient.
→Thank you for this comment. We have added them to the text.
Did any patient take calcium or vitamin D supplementation before or during the observation period?
→Yes, this has been added to the text.
The authors should specify how the fracture events were collected in the methods section.
→We have added this point in the Methods section.
The dosage of calcium and phosphorus should be included in the method section. Calcium levels, such as all laboratory variables, should better be expressed in SI units.
→Thank you for this suggestion. We have added the dosage of calcium and phosphorus in the Methods section. We apologize that the unit of calcium in the submitted manuscript was incorrect. We have changed it from “mEq/L” to “mg/dL”.
Figure 2: In Patient 3 and not 5, the percent changes in urinary NTX sharply increased at 30 months
→Thank you for pointing this out. We have corrected this in the Legend.
For patient 5, the timing of the incident fracture should be specified
→We have added this point in the text.
For Patient 7, all laboratory data are notified until 6M except calcium. Is that a mistake?
→We apologize for this oversight. We have edited the Figure.
The evolution of biological and BMD individual data is difficult to interpret without the timing of denosumab administration for each patient.
→Actually, we synchronized the timing period of BMD measurements after the start of denosumab therapy for each patient in this study. We have mentioned this in the text.
The major concern for denosumab therapy is the reverse effect after discontinuation and the occurrence of fractures. Did the authors follow those patients after this study? Did they have information about subsequent treatments or fractures events?
→We agree with this concern and have mentioned it in the Discussion. Unfortunately, we do not have precise data on our cohort after discontinuation, but will address this point in a future study.
#Reviewer 3
Major criticisms:
The study group is very small (considering also that there are two mother+ daughter couples) and heterogeneous: 7 females ranging from early childhood to post-menopause, 1 adult male.
→ We completely agree with this point. However, this study is admittedly submitted as a case series and not an original article. The small cohort size has also been stated as a study limitation.
The causative mutations have not been reported precisely anywhere in the paper (they should be added in a column in Table 1). In Results, lanes 102-105, four different (?) collagen I mutations are just mentioned aspecifically; no COL I mutation has been found in two patients (# 2 and #3), yet they are classified as OI type I. Type I OI, the most common and mildest form of Osteogenesis Imperfecta, has been always associated to nonsense/missense/frameshift/ mutations in collagen type I genes (COL1A1 in most cases, COL1A2 in fewer cases). There are two possibilities: either the mutations were missed for technical reasons or patients #2 and #3 have a different OI type, as the authors hypotesize in the Discussion (lanes 240-241). A more accurate mutation screening, including SERPINF1 and IFITM5 genes, for instance, should be done. This is an important issue, in my opinion, since it is known, that OI type VI patients (disease gene=SERPINF1) and OI type V patients (disease gene= IFITM5) respond poorly to BPBs treatment, while they seem to respond well to Denosumab treatment. The clinical phenotypes of these two forms may be relatively mild and therefore be misclassified as type I.
→Thank you very much for these excellent comments. We have added 5 co-authors (human geneticists) who performed additional next generation sequencing in this study. They confirmed that there were no remarkable mutations in Patients 2 or 3, including COL1A1, COL1A2, SERPINF1, and IFITM5 genes. However, their clinical phenotypes were type I similar. We have added these points in the text. We have also added as much information as possible in Table 1. We understand and appreciate the Reviewer’s queries and worries, and offer our best reply.
-2. Denosumab decreases serum calcium levels in the first period after the injection as a consequence of the inhibition of bone turnover. In this case series, calcium levels were unchanged in almost all patients. How do the authors explain this difference? Were the patients taking calcium and/or vitamin D? At what dosage?
→Thank you for this observation. Actually, we earlier observed that serum calcium levels could remain constant even without vitamin D or calcium supplementation (Nakamura et al., Bone Res. 2017 Oct 10;5:17021.). We have added this point in the text as well as information on the patients who had received supplementation.
-3. In this small number of cases the high range of variation in bone turnover markers makes results difficult to interpret. Why did the authors choose urinary NTX and TRACP-5b as resorption markers instead of serum CTX, that is more accurate?
→We appreciate this comment. CTX is indeed an established resorption marker. In addition, numerous studies have reported that urinary NTX and TRACP-5b are good bone resorption markers as well (Yamamoto et al. J Bone Miner Metab. 2014 Nov;32(6):699-708.; Miyagi et al. J Orthop Sci. 2018 Aug 23. pii: S0949-2658(18)30224-0.). Based on these, we believe that our data are also reliable.
4. The high number of densitometric evaluations with a small time interval is unnecessary and questionable.
→Thank you very much for this comment. As suggested, it would have been possible to reduce the frequency of densitometric measurements in the manuscript. However, since there are few reports on Japanese osteoporotic OI patients with denosumab therapy, we believed that a small time interval would be useful for experts in this field, particularly in Japan. If the Reviewer recommends the data be deleted, please notify us.
5. The dosage of 25-OH vitamin D is considered the gold standard for monitoring vitamin D level. Why did the authors evaluate only 1,25 OH2 vitamin D?
→ We absolutely agree with this comment. In Japan, however, only a single measurement of 25(OH)D is allowed for each patient during osteoporosis treatment, so we could not include these data in the present study. We will add such measurements in the future. We have stated this point as a limitation.
6. Previous papers have described the positive effects of Denosumab treatment in young and adult OI patients (eg: Semler O. et al 2012, J Musc Neuronal Interact; Trejo P. et al 2018 J Musc Neuronal Interact; Hoier-Kuhn H et al 2016, J Musc Neuronal Interact; Hoier-Kuhn H et al Pediatr Endocrinol Rev. 2017 Nov;15; Uehara M, et al 2017 Tohoku J Exp Med. 2017), therefore I do not find any particular novelty in the present study .
→ We understand the Reviewer’s perspective. To the best of our knowledge, however, there has been only one such report in Japan, which was published by our group. In that previous paper, we described 3 cases of OI over a 2-year observational period. In contrast, the present manuscript presents 8 cases over 56 months. Thus, we believe our report contains important novelty and is worthy of publication.
Minor criticisms
1. Abstract, line 14: add an a after “imperfect”
→We have added this, thank you.
2. Introduction, lines 36-38: the sentence “Cartilage-associated protein………Morello et al. 2016” is misplaced, it should be erased.
→Thank you. We have erased it.
The different OI disease genes should be discussed collectively in the Discussion, starting from lane 227.
→As suggested, we have collectively discussed these parts in the Discussion.
3. Results, line 106: “Patients 1 and 4 had no family history of OI”. Patient 1 is indicated as being the mother of patient 5: it would be more appropriate to consider her as the first mutant in a familial case of OI type I.
→We apologize for this oversight and have deleted “Patients 1 and 4 had no family history of OI”.
4. The text may be improved if revised by a native English speaker
→ We have had the paper rechecked by a native English speaker to improve the text. Several additions and deletions made in the editing process were not highlighted in the revised version..
Reviewer 2 Report
The authors have retrospectively evaluated the effect of denosumab on several bone markers, bone mineral density and fractures in 8 cases of child and adult osteogenesis imperfecta with osteoporosis.
They showed an inhibition of BTM and an increased of BMD in most of patients, a low fracture rate and no severe side effects.
Specific comments:
The duration, dose and frequency of Denosumab administration should be specified for each patient.
Did any patient take calcium or vitamin D supplementation before or during the observation period?
The authors should specify how the fracture events were collected in the methods section.
The dosage of calcium and phosphorus should be included in the method section. Calcium levels, such as all laboratory variables, should better be expressed in SI units.
Figure 2: In Patient 3 and not 5, the percent changes in urinary NTX sharply increased at 30 months
For patient 5, the timing of the incident fracture should be specified
For Patient 7, all laboratory data are notified until 6M except calcium. Is that a mistake?
The evolution of biological and BMD individual data is difficult to interpret without the timing of denosumab administration for each patient.
The major concern for denosumab therapy is the reverse effect after discontinuation and the occurrence of fractures. Did the authors follow those patients after this study? Did they have information about subsequent treatments or fractures events?
Author Response

(The authors gave the same response as above.)

Reviewer 3 Report
The manuscript by Kobayashi et al, describes the effects of long term (>30 mo) Denosumab therapy on a small group (8 subjects, age range: 3-51) of OI patients. The clinical outcomes reported are: no averse effects during denosumab treatment, BMD increase in almost all cases, only 1 fracture reported in 1 pt during treatment, despite 6 out of 8 pts had suffered >10 fractures before the study. The authors conclude that Denosumab may represent a good therapeutic option for OI with osteoporosis.
Major criticisms:
-1. The study group is very small (considering also that there are two mother+ daughter couples) and heterogeneous: 7 females ranging from early childhood to post-menopause, 1 adult male. The causative mutations have not been reported precisely anywhere in the paper (they should be added in a column in Table 1). In Results, lanes 102-105, four different (?) collagen I mutations are just mentioned aspecifically; no COL I mutation has been found in two patients (# 2 and #3), yet they are classified as OI type I. Type I OI, the most common and mildest form of Osteogenesis Imperfecta, has been always associated to nonsense/missense/frameshift/ mutations in collagen type I genes (COL1A1 in most cases, COL1A2 in fewer cases). There are two possibilities: either the mutations were missed for technical reasons or patients #2 and #3 have a different OI type, as the authors hypotesize in the Discussion (lanes 240-241). A more accurate mutation screening, including SERPINF1 and IFITM5 genes, for instance, should be done. This is an important issue, in my opinion, since it is known, that OI type VI patients (disease gene=SERPINF1) and OI type V patients (disease gene= IFITM5) respond poorly to BPBs treatment, while they seem to respond well to Denosumab treatment. The clinical phenotypes of these two forms may be relatively mild and therefore be misclassified as type I.
-2. Denosumab decreases serum calcium levels in the first period after the injection as a consequence of the inhibition of bone turnover. In this case series, calcium levels were unchanged in almost all patients. How do the authors explain this difference? Were the patients taking calcium and/or vitamin D? At what dosage?
-3. In this small number of cases the high range of variation in bone turnover markers makes results difficult to interpret. Why did the authors chooseurinary NTX and TRACP-5b as resorption markers instead of serum CTX, that is more accurate?
-4. The high number of densitometric evaluations with a small time interval is unnecessary and questionable.
-5. The dosage of 25-OH vitamin D is considered the gold standard for monitoring vitamin D level. Why did the authors evaluate only 1,25 OH2 vitamin D?
-6. Previous papers have described the positive effects of Denosumab treatment in young and adult OI patients (eg: Semler O. et al 2012, J Musc Neuronal Interact; Trejo P. et al 2018 J Musc Neuronal Interact; Hoier-Kuhn H et al 2016, J Musc Neuronal Interact; Hoier-Kuhn H et al Pediatr Endocrinol Rev. 2017 Nov;15; Uehara M, et al 2017 Tohoku J Exp Med. 2017), therefore I do not find any particular novelty in the present study .
Minor criticisms
1. Abstract, line 14: add an a after “imperfect”
2. Introduction, lines 36-38: the sentence “Cartilage-associated protein………Morello et al. 2016” is misplaced, it should be erased. The different OI disease genes should be discussed collectively in the Discussion, starting from lane 227.
3. Results, line 106: “Patients 1 and 4 had no family history of OI”. Patient 1 is indicated as being the mother of patient 5: it would be more appropriate to consider her as the first mutant in a familial case of OI type I.
4. The text may be improved if revised by a native English speaker
Author Response

(The authors gave the same response as above.)

Round 2
Reviewer 2 Report
The authors have satisfactorily answered to most of questions except the first one. Indeed the duration and the frequency of Denosumab administration should be specified for each patient in Tables. In the method section the authors indicate that Denosumab was injected every 6 months but patients were not all followed until 54 months and 33 is not a multiple of 6.. Moreover in Table 2 are the times 2-4M, 27M, 33M, … correspond to Denosumab administration?
Author Response
Editor-in-Chief
Journal of Clinical Medicine
November 15, 2018
Dear Editor-in-Chief,
Please find enclosed our revised manuscript, entitled “Efficacy and Safety of Denosumab Therapy for Osteogenesis Imperfecta Patients with Osteoporosis - Case Series”, which we would like to have re-considered for publication in Journal of Clinical Medicine. We have addressed the remaining comments raised by the Referee to the best of our ability.
Thank you again for your time and consideration.
Sincerely,
Yukio Nakamura
#2 Reviewer
The authors have satisfactorily answered to most of questions except the first one. Indeed the duration and the frequency of Denosumab administration should be specified for each patient in Tables.
→Thank you for this advice. We have added the information to Table 1.
In the method section the authors indicate that Denosumab was injected every 6 months but patients were not all followed until 54 months and 33 is not a multiple of 6..
→Thank you for raising this concern. In all patients, denosumab was injected every 6 months. We investigated bone turnover markers and bone mineral density at different periods for each patient since we wanted to collect as much data as possible prior to submission.
We have added the following description to clarify these points:
“Subcutaneous denosumab injections of 60 mg in Patients 1, 2, 3, 4, 6, and 7, 30 mg in Patient 8, and 15 mg in Patient 5 were administered every 6 months. Patient data at different follow-up periods (i.e., 0, 6, 12, 18, 24, 30, 33, 39, 42, 49, and 54 months) were included to maximize analytical power.”
Moreover in Table 2 are the times 2-4M, 27M, 33M, … correspond to Denosumab administration?
→We appreciate this question and have added the following in the titles of Tables 2-4: “…across observation periods for each patient”.

Reviewer 3 Report
The authors have accepted most of the suggestions made in my first report. They have made an effort to improve the manuscript’s quality.
Therefore I think that the revised version may be accepted for publication
Author Response
Editor-in-Chief
Journal of Clinical Medicine
November 15, 2018
Dear Editor-in-Chief,
Please find enclosed our revised manuscript, entitled “Efficacy and Safety of Denosumab Therapy for Osteogenesis Imperfecta Patients with Osteoporosis - Case Series”, which we would like to have re-considered for publication in Journal of Clinical Medicine. We have addressed the remaining comments raised by the Referee to the best of our ability.
Thank you again for your time and consideration.
Sincerely,
Yukio Nakamura
#2 Reviewer
The authors have satisfactorily answered to most of questions except the first one. Indeed the duration and the frequency of Denosumab administration should be specified for each patient in Tables.
→Thank you for this advice. We have added the information to Table 1.
In the method section the authors indicate that Denosumab was injected every 6 months but patients were not all followed until 54 months and 33 is not a multiple of 6..
→Thank you for raising this concern. In all patients, denosumab was injected every 6 months. We investigated bone turnover markers and bone mineral density at different periods for each patient since we wanted to collect as much data as possible prior to submission.
We have added the following description to clarify these points:
“Subcutaneous denosumab injections of 60 mg in Patients 1, 2, 3, 4, 6, and 7, 30 mg in Patient 8, and 15 mg in Patient 5 were administered every 6 months. Patient data at different follow-up periods (i.e., 0, 6, 12, 18, 24, 30, 33, 39, 42, 49, and 54 months) were included to maximize analytical power.”
Moreover in Table 2 are the times 2-4M, 27M, 33M, … correspond to Denosumab administration?
→We appreciate this question and have added the following in the titles of Tables 2-4: “…across observation periods for each patient”.
#2 Reviewer
Thank you very much.
